# Enhancer-Mediated Formation of Nuclear Transcription Initiation Domains

**DOI:** 10.3390/ijms23169290

**Published:** 2022-08-18

**Authors:** Matthew D. Gibbons, Yu Fang, Austin P. Spicola, Niko Linzer, Stephen M. Jones, Breanna R. Johnson, Lu Li, Mingyi Xie, Jörg Bungert

**Affiliations:** Department of Biochemistry and Molecular Biology, Center for Epigenetics, Genetics Institute, UF Health Cancer Center, Powell-Gene Therapy Center, University of Florida, Gainesville, FL 32610, USA

**Keywords:** RNA polymerase II, transcription, enhancer, super-enhancer, eRNA, transcription initiation domain

## Abstract

Enhancers in higher eukaryotes and upstream activating sequences (UASs) in yeast have been shown to recruit components of the RNA polymerase II (Pol II) transcription machinery. At least a fraction of Pol II recruited to enhancers in higher eukaryotes initiates transcription and generates enhancer RNA (eRNA). In contrast, UASs in yeast do not recruit transcription factor TFIIH, which is required for transcription initiation. For both yeast and mammalian systems, it was shown that Pol II is transferred from enhancers/UASs to promoters. We propose that there are two modes of Pol II recruitment to enhancers in higher eukaryotes. Pol II complexes that generate eRNAs are recruited via TFIID, similar to mechanisms operating at promoters. This may involve the binding of TFIID to acetylated nucleosomes flanking the enhancer. The resulting eRNA, together with enhancer-bound transcription factors and co-regulators, contributes to the second mode of Pol II recruitment through the formation of a transcription initiation domain. Transient contacts with target genes, governed by proteins and RNA, lead to the transfer of Pol II from enhancers to TFIID-bound promoters.

## 1. Introduction

The first evidence of transcription at or through enhancers came from studies of Hox, globin, and B-cell-specific gene loci [1,2,3,4,5,6,7,8]. These early studies suggested that transcription through enhancers could change their epigenetic state, or that transcription initiated at enhancers could deliver a transcription complex to target promoters through looping or tracking mechanisms [1,2,5,6,7,8]. More current studies show that RNA polymerase II (Pol II) is recruited to many, if not most, enhancers and initiates the formation of noncoding enhancer RNA (eRNA) [9,10]. Recently, it was shown that Pol II and other components of the transcription complex are recruited to upstream activating sequences (UASs) in yeast, suggesting that Pol II recruitment to distal regulatory elements is an evolutionarily conserved mechanism in eukaryotes [11]. However, it is still a matter of investigation whether enhancers represent the primary site of transcription complex recruitment, and whether enhancer-associated eRNA functionally contributes to enhancer-mediated transcription activation at promoters. In the following sections, we summarize the knowledge on the formation of Pol II transcription complexes at promoters as well as at enhancers/super-enhancers, and we then present what is known about the potential functional role of enhancer transcription and eRNA. This review focuses on activities and mechanisms that recruit Pol II to enhancers and establish transient contacts with promoters.

## 2. Promoter and Formation of a Pol II Transcription Complex

Pol II is recruited to basal promoter elements with the help of general transcription factors (GTFs) [12]. Among the basal promoter elements are the TATA box, the Initiator, and the Downstream Promoter Element (DPE). These sequences are bound by components of the transcription factor TFIID complex [13,14]. The TATA-binding protein (TBP) interacts with the TATA box, while TBP-associated factors 1 and 2 (TAF1 and TAF2) interact with the Initiator and DPE. The binding of TBP to the TATA box is stabilized by TFIIA. The TFIID complex exists in different conformations, e.g., canonical and rearranged [13]. In the canonical configuration, TBP does not interact with the TATA box. The rearranged configuration exposes the TBP DNA-binding domain. The overall mode of binding of TFIID to TATA-containing and TATA-less promoters is similar [14]. This is relevant as most Pol II transcribed genes do not contain a consensus TATA sequence.

The recruitment of Pol II and associated TFIIF to promoter-bound TFIID is mediated by TFIIB [12]. Pol II contains a long unstructured C-terminal domain (CTD) harboring multiple repeats of a heptapeptide that is subject to phosphorylation at different stages of the transcription cycle [15]. The unphosphorylated form of Pol II is recruited to the TFIID/TFIIB complex at the promoter, which is followed by the binding of TFIIE. The association of TFIIH, mediated by TFIIE, completes the formation of the Pol II transcription preinitiation complex. TFIIH promotes DNA strand separation and phosphorylates serine 5 of the CTD, which disrupts contacts with the basal transcription factors and the co-regulator complex Mediator. Transcription initiation is paused close to the transcription start site (TSS) to allow capping of the 5′ end of the mRNA. For transcription elongation to continue, pTEFb (positive transcription elongation factor b) phosphorylates negative elongation factors NELF and DSIF, as well as the serine 2 residues of the Pol II CTD repeats [16]. This leads to the displacement of NELF, the conversion of DSIF into a positive transcription elongation factor, and the recruitment of additional proteins that stimulate transcription elongation.

The GTFs and Pol II are sufficient to initiate transcription of a TATA-box-containing promoter in the test tube. In the cellular context, DNA sequence-specific transcription factors and co-regulators are required for efficient transcription by Pol II. The co-regulators act by displacing nucleosomes, modifying histone tails, or facilitating recruitment of GTFs and transcription elongation factors. Of note is the acetylation and methylation of nucleosomes immediately downstream of the TSS. Acetylated nucleosomes are recognized by TAF1, and methylated nucleosomes (H3K4me3) interact with TAF3 [17]. These interactions may represent the first event in the association of TFIID with a promoter in vivo.

## 3. Enhancers and Super-Enhancers

Enhancers are DNA sequences that positively regulate the transcription of target genes independent of distance, position, and orientation [18]. These elements contain several binding sites for DNA sequence-specific transcription factors, which associate with co-regulators, including chromatin-modifying protein complexes, CBP/p300, Mediator, and Brd4 [19,20,21,22]. The co-regulatory activities increase accessibility at specific sites in the genome and stimulate the recruitment and/or elongation of transcription complexes. Often, genes are regulated by multiple enhancer elements that may act additively or synergistically.

Active enhancers, like promoters, are sensitive to digestion by DNaseI, referred to as hypersensitive sites (HSs) [23]. The accessibility to nucleases is due to the displacement of nucleosomes or to the incorporation of histone variants that render the DNA more accessible.

Super-enhancers (SEs) are extended genomic regions often containing multiple separated enhancer elements that function together to mediate extremely high-level expression of target genes [24,25,26]. These complex elements are characterized by abundant association with Mediator, as well as Pol II, and extended regions of elevated levels of histone 3 lysine 27 acetylation (H3K27ac) and H3K4 mono-methylation (H3K4me) [27,28]. The acetylation of H3K27 is catalyzed by p300/CBP, which is recruited to enhancers by RNA and sequence-specific transcription factors [20,29]. Both TAF1, a component of TFIID, and Brd4 recognize acetylated histones and may thus be recruited to sites flanking the enhancers, from where they stimulate transcription initiation and elongation, respectively [30,31].

Enhancers activate transcription by different mechanisms, including modifying chromatin structure, relocating gene loci in the nucleus, or directly assisting in the recruitment of transcription complexes. Most current studies suggest that enhancers come in close proximity to target genes to modulate transcription efficiency [19]. The proximity of the enhancer and promoter could facilitate the recruitment of components of the transcription complex and/or could stimulate transcription elongation of a paused Pol II. How close enhancers and promoters come together is a matter of debate. Chromosome conformation capture (3C) experiments demonstrate that enhancers and promoters interact more frequently when genes are activated [32]. Enhancer/promoter contacts could be established via interactions between proteins interacting with these elements. This model has been supported by data showing that the artificial tethering of a protein dimerization domain to a globin promoter recruits the enhancer and stimulates transcription [33]. However, a study from the Bickmore laboratory indicates that upon activation of transcription, the distance between a promoter and enhancer increased [34]. This may be explained by studies suggesting that enhancers form phase-separated transcription initiation domains [27,28]. Both enhancers and promoters could be at the periphery of these domains, and, depending on the size, distances between enhancers and promoters could be increased upon activation.

## 4. Transcription Complex Recruitment at Enhancers, Super-Enhancers, and Yeast Upstream Activating Sequences

It has been known for some time that enhancers recruit Pol II and other GTFs [5,6]. However, it was not recognized until the 2010s that the recruitment of Pol II at enhancers is widespread [9]. Early work on the HoxD4, lambda5/VpreB1, and globin gene loci detected Pol II and GTFs at enhancers regulating one or multiple genes. It was postulated that transcription complexes are first recruited to these enhancers and then transferred to target genes by tracking or looping mechanisms [2,5,7,8]. The transfer of Pol II from enhancers to promoters could be facilitated by the presence of strong basal promoter elements at target genes.

A recent study demonstrated that Pol II, TFIIF, and TFIIE are recruited to upstream activating sequences (UASs) in yeast [11] (Figure 1A). This suggests that the recruitment of Pol II by transcription factors and co-regulators at distal regulatory elements is a conserved process. Interestingly, TFIIH was recruited at the basal promoter but not at UASs [11]. This suggests that association of TFIIH is linked to TFIID/TFIIB-mediated recruitment of Pol II at basal promoter elements.

How transcription complexes are recruited to enhancers in higher eukaryotes is not completely understood. It is possible that similar mechanisms operating at promoters, including the placement of a TFIID complex, also mediate the recruitment of Pol II to enhancers. This may involve the recruitment of TFIID, through TAF1, to H3K27 acetylated nucleosomes flanking enhancers and the subsequent recruitment of PolII, TFIIF, and TFIIE by TFIIB (Figure 1B). Association of TFIIH renders Pol II transcription competent, leading to the formation of eRNA. Additional Pol II may be recruited by transcription factors via Mediator and other co-activators, leading to a high concentration of Pol II at enhancers and SEs, which would then be transferred to TFIID-bound promoters during transient contacts [11,21,27,28,35] (Figure 1B and Figure 2).

It is possible that most Pol II complexes recruited to enhancers/super-enhancers will not engage with DNA to initiate transcription. As a consequence, recruitment of Pol II via Mediator or other proteins may not be efficiently cross-linked to DNA in ChIP assays, thus potentially underestimating the amount of Pol II at SEs. 

In summary, there appear to be two modes of Pol II recruitment to enhancers and SEs. One mechanism involves the acetylation of H3K27 by CBP/p300, followed by the recruitment of TFIID via TAF1 and Pol II/TFIIF/TFIIE via TFIIB. This leads to the recruitment of TFIIH and initiation of eRNA transcription. The other mode involves the recruitment of Pol II/TFIIF/TFIIE via transcription factors and co-activators [11]. The first mode of recruitment leads to the generation of eRNA, and the second mode provides a dynamic pool of Pol II that is rendered transcription competent after transfer to TFIID-bound promoters (Figure 1B and Figure 2).

## 5. Enhancer-Mediated Transcription Bursting

Transcription at many genes occurs in bursts, which are characterized by burst size and frequency [36,37,38]. The burst size is determined by basal promoter elements and Mediator, while frequency is determined by enhancer elements. The bursting phenomena are consistent with transient interactions between the enhancer and promoter during which either Pol II is recruited to the promoter or a paused Pol II is converted to an elongating Pol II. It should be noted that transcription bursting in yeast occurs independently of enhancer-mediated activation, and it is determined by the binding of transcription activators [39]. Recently, it was shown that accumulation of mRNA disrupts the transcription initiation domains at SEs [38,40]. Furthermore, the Felsenfeld group demonstrated that an intronic sequence in a globin mRNA interacts with one of the β-globin locus control region (LCR) SE-associated HS sites via triple helix formation and blocks the binding of transcription factors [41]. Thus, the transcription bursting at highly expressed genes could, in part, represent a feedback control that prevents transcription initiation mediated by the enhancer/SE transcription initiation domain in order to allow ordered processing and nuclear export of the accumulating mRNA [40].

## 6. The Role of Enhancer Transcription and eRNA

Transcription at enhancers occurs in a cell-type-specific and mostly bidirectional manner [42]. The eRNAs are 5′ capped but in most cases are neither spliced nor poly-adenylated [43]. This may be due to the overall under-representation of U1 splice sites and rapid transcription termination [44]. However, noncoding RNAs, including eRNAs, are enriched for 5′ splice sites and depleted of 3′ splice sites. Associations with the U1snRNP complex mediate the retention of noncoding RNA with chromatin in a transcription dependent manner [45]. Studies in Drosophila have shown that the half-life of paused Pol II is shorter at enhancers compared to promoters [46]. Although eRNAs are generally less stable than mRNAs, their average half-life ranges from 5 to 10 min. The relatively short half-life suggests that eRNA function is transient [47,48].

The observation of bidirectional transcription at enhancers is restricted to the population of cells. At the single cell level, it appears that divergent eRNA transcripts rarely co-localize [49]. Moreover, a recent study suggests that eRNA transcription does not correlate with enhancer-mediated transcription activation of target genes. It is thus possible that at certain enhancers, transcription and/or eRNA function is restricted to the early stages of enhancer complex formation [50].

It has been known for a while that the process of transcription alters chromatin structure. More recently, accumulating evidence suggests that eRNAs play functional roles in enhancer-mediated activation of transcription. The eRNA could function in a sequence-independent manner by participating in the establishment of a transcription initiation domain, or it could function in a sequence-dependent manner by interacting with proteins, RNA, or DNA. In the following, we discuss how enhancer transcription and/or the resulting eRNA may contribute to Pol II recruitment at enhancers and the transfer of Pol II from enhancers to promoters.

**The role of transcription initiated at enhancers/SEs**: There are several mechanisms by which enhancer transcription could impact transcription activation. For example, the process of transcription at enhancers modifies flanking nucleosomes, and the modified nucleosomes recruit proteins important for enhancer function. Several histone marks are primarily found at enhancer regions, including H3K27ac (histone H3 lysine 27 acetylation) and H3K4me (H3K4 monomethylation) [51]. These modifications are introduced by histone-modifying enzymes recruited to enhancers through DNA sequence-specific transcription factors. For example, the histone acetyltransferase (HAT) p300/CBP interacts with many different transcription factors and acetylates H3K27. This histone modification assists in the recruitment of Brd4 and pTEFb, leading to the stimulation of transcription elongation [52].

Other histone modifications are introduced during transcription and depend in part on the phosphorylation status of the CTD [53]. For example, in yeast, the Compass complex, which contains the H3K4 histone methyltransferase (HMT) Set1, is recruited to the serine 5 phosphorylated CTD. This correlates with H3K4me3 peaks which localize just downstream of TSSs [54]. In higher eukaryotes, the situation is more complex and involves different H3K4 HMTs [53]. The activity of these HMTs is stimulated by H2B ubiquitination (H2Bub). The H2Bub enzymes are recruited to the transcribing Pol II through the PAF1 complex, a multi-subunit protein complex that associates with the transcribing Pol II after dissociation of NELF [55]. H3K36me is introduced during transcription by HMTs that interact with the serine 2 phosphorylated form of the CTD. With regard to enhancer function, it is interesting to note that H3K4me and H3K36me stabilize H3K27 acetylation by preventing the recruitment of PRC (polycomb repressive complex), which mediates H3K27 methylation [56]. Thus, the process of transcription at enhancers could stabilize the H3K27ac mark and, as a consequence, promote the recruitment of TAF1, Brd4 and pTEFb.

Noncoding transcription has also been associated with chromatin conformational changes, thereby facilitating promoter–enhancer interactions. Topoisomerases associate with the transcribing Pol II and change the topology and folding of DNA. Jha et al. [57] proposed that transcription-driven positive supercoiling could compact DNA and decrease the distance between enhancer and promoter. Furthermore, several transcription factors were shown to bind more efficiently to DNA when the binding sequence was in the context of negative supercoils, which accumulate behind the transcribing polymerase [58]. This could be relevant to bidirectional transcription initiating within enhancers. Even though it was shown that most enhancers are transcribed unidirectionally at a given time, it is possible that rare and transient bidirectional transcription, promoted by R-loops, leads to the ordered and stable association of protein complexes at enhancers [59]. Divergent transcription is further stimulated by RNA exosome-mediated removal of R-loops [60].

**The role of eRNA in the formation of enhancer-associated transcription initiation domains**: The nucleus contains membraneless organelles (MLOs) including the nucleolus where rRNA transcription and ribosome assembly take place; histone bodies, which are sites of histone gene transcription; and nuclear speckles, which are specialized for mRNA processing [61]. Several studies have implicated the formation of noncoding RNA in the seeding of MLOs [61]. Super-enhancers may also form phase-separated MLOs specialized for efficient transcription initiation at target genes [27,28]. Moreover, there is evidence that eRNA participates in the formation of SE transcription initiation domains [62]. Phase separation is often mediated through multi-valent interactions between intrinsically disordered domains of proteins, between proteins and RNA, and between different RNA species, which could be direct or mediated by RNA binding proteins [61,63]. A recent systematic study on RNA length, protein–RNA stoichiometry, and the ionic environment revealed that the length of the RNA regulates the surface tension and stability of RNA–protein condensates [64].

There is still some debate about the physical nature of SE transcription initiation domains. A recent study demonstrated that interactions between transcription factors through intrinsically disordered activation domains enhanced transcription by increasing DNA residence time in the absence of phase separation [65]. Whatever the physical nature of SE transcription initiation domains, eRNA may assist in the concentration and/or compartmentalization of activities required for efficient target gene expression.

**Interactions of eRNA with proteins**: eRNAs have been shown to interact with transcription factors, co-regulators, and proteins involved in RNA metabolism [42], and the presence of eRNA at specific genomic loci correlates with transcription factor activity [66]. Furthermore, eRNAs have been shown to sequester negative co-regulators, including chromatin-modifying complexes and negative elongation factors (e.g., NELF) [42,67]. This alters chromatin accessibility or transcription elongation. Likewise, interactions with CBP/p300, Brd4, and pTEFb positively enhance transcription through increasing chromatin accessibility, stimulating the recruitment of basal transcription factors to promoters, and promoting transcription elongation [52].

As mentioned previously, Pol II is paused by negative elongation factors DSIF and NELF [16]. Phosphorylation of these proteins by pTEFb dissociates NELF and converts DSIF into a positive transcription elongation factor. It was shown that eRNA of a length greater than 200 nucleotides and containing unpaired guanosines interacts with NELF subunits allosterically and causes release from Pol II, thereby stimulating transcription elongation [68]. Thus, enhancers may contribute to efficient transcription elongation by two distinct mechanisms: the p300/CBP mediated acetylation of H3K27, which recruits Brd4 and, consequently, pTEFb, and eRNA-mediated dissociation of NELF.

Hou and Kraus targeted eRNAs to estrogen receptor α (ERα) enhancers and found that specific eRNAs carrying a 40-nucleotide motif called FERM (functional eRNA regulatory motif) enhanced binding of ERα and elevated levels of H3K27ac [69]. Proteomic studies identified SPF27 (BCAS2), a spliceosome component, as interacting with FERM. In support of these findings, SPF27 has previously been shown to co-regulate ERα dependent transcription [70]. Importantly, the FERM motif was found in a large number of eRNAs [69].

**Interaction of eRNA with RNA and DNA**: There is increasing evidence that noncoding RNA interacts with DNA, either in the context of R-loops or in the context of RNA:DNA triple helices [71,72]. For example, a specific eRNA in the protocadherin (Pcdh) α HS1-5 enhancer region forms an R-loop within the enhancer and brings the target promoter in close proximity [73]. Furthermore, RNA, including eRNA, can form triple helices with double-stranded DNA via Hoogsteen base pairing (Triplexes) [72]. The RNA in these triplexes can bind in *cis* or *trans* and often recruits co-regulators, including the PRC complex to repress or p300/CBP to activate transcription. For example, the SARRAH lncRNA acts in *trans* to recruit p300/CBP to several cardiac-specific genes, resulting in elevated H3K27ac levels [74]. In addition to these documented cases of co-regulator recruitment by triplexes, segments of eRNAs, perhaps still tethered to the enhancer or part of a phase-separated transcription initiation domain at SEs, could interact with specific DNA sequences in the proximity of target genes, thereby promoting transient contacts between enhancers and promoters.

Recently, several studies have shown interactions of eRNA with transcripts generated at TSSs in an antisense direction (promoter upstream transcripts, PROMPTs) [75]. For example, interactions between eRNA and PROMPTs, mediated by RNA-binding proteins, facilitate contacts of the Myc gene with a distantly located SE [76]. A similar situation was found in the HoxDmiR-10b gene cluster in which a promoter-associated antisense (AS) transcript (HoxD-AS) and an enhancer-associated lncRNA (LNC01116) interact to mediate proximity between these regulatory elements [77].

In summary, there is evidence that the transcription at enhancers changes the chromatin configuration and DNA topology, which may stimulate the binding of transcription factors and co-regulators and decrease the distance between enhancers and promoters. Moreover, there is evidence for a functional role of eRNA either through the formation of enhancer-centered transcription initiation domains, through the recruitment of positive and removal of negative regulators, or through interactions with RNA and DNA, thereby promoting transient enhancer–promoter contacts. Thus, increasing evidence suggests that transcription at enhancers and the resulting eRNAs participate in the formation of transcription initiation domains that concentrate Pol II and also contribute to the transfer of Pol II from the initiation domain to promoters by facilitating promoter–enhancer interactions.

## 7. Establishment of Promoter–Enhancer Contacts

It is still unclear how enhancers select target genes in the genome. Transcription factors, like YY1, and cofactors, like Ldb1, have been shown to mediate proximity between specific enhancers and promoters via direct protein–protein interactions [33,78]. How could this be relevant in the context of transcription initiation domains? El Khattabi et al. [79] provided evidence that transcription factors may diffuse from the enhancers and contact promoters to activate transcription. Thus, it seems possible that transcription factors and co-regulators are first recruited to enhancers and SEs to create a transcription initiation domain that concentrates Pol II and other components of the transcription complex. Proteins like Ldb1, associated with transcription factors, together with eRNA may diffuse to the surface of these domains and establish transient contacts with genes so that Pol II can be transferred to high-affinity basal promoter elements (Figure 2).

In addition to the aforementioned mechanisms, CTCF (CCCTC-binding factor) and cohesin are involved in organizing the genome in a way that facilitates promoter–enhancer interactions [80]. It is thus interesting to note that the lncRNA HOTTIP forms R-loops near CTCF binding sites, interacts with CTCF, and promotes the formation of chromosomal loops [81].

## 8. Conclusions and Outlook

Enhancers and promoters function in a specific chromosome environment and in a specific context with regard to distance and relative position. Often, genes are regulated by multiple enhancers that may drive expression in a cell-type- or developmental-stage-specific manner. Many highly expressed genes are regulated by SEs in which multiple enhancers act together to form a transcription initiation domain that concentrates eRNA, co-regulators, and Pol II. In the context of transcription initiation domains, Mediator and other co-regulators recruit a large pool of transcription-incompetent Pol II, which is transferred to TFIID-bound promoters during transient contacts. A small fraction of Pol II recruited to enhancers or SEs is used to generate eRNA (Figure 1B and Figure 2). The recruitment of transcribing Pol II at enhancers likely involves mechanisms similar to those operating at promoter regions. Whether it is the process of transcription and/or the eRNA transcript that contributes to the activation of target genes is a matter of ongoing investigation. Nevertheless, there is increasing evidence for a functional role of the transcription process, through either the modulation of chromatin structure, the formation of R-loops, or the release of transcription-competent Pol II that is transferred to target gene promoters. There is also increasing evidence for a functional role of eRNA, which involves the formation of a phase-separated transcription initiation domain, interactions with proteins that mediate transcription initiation and elongation at target genes, or bridging the distance between enhancers and promoters. We suggest that the eRNA is involved in the formation of a transcription initiation domain that concentrates Pol II at enhancers and, in addition, participates together with proteins in establishing transient contacts with promoters via triplexes and/or via interactions with PROMPTs (Figure 2). The concept of enhancer-centered transcription initiation domains is similar to the transcription factory model proposed by Cook and colleagues [82]. The transcription factory model proposed that Pol II forms clusters in the nucleus to which genes are recruited and reeled through during transcription. In contrast to the transcription factory model, enhancer-centered transcription initiation domains form in a DNA sequence-specific manner, transiently associate with genes, and transfer Pol II to promoters. Pol II then elongates away from the initiation domain.

Improved new technologies for mapping RNA–RNA, RNA–DNA, RNA–protein, and DNA–protein interactions genome-wide, as well as for identifying enhancer and SE-specific protein complexes, will lead to further progress in our understanding of long-distance gene regulation. Furthermore, there has been enormous progress in high-resolution imaging, which will further illuminate the architecture of enhancers and promoters in the context of specific nuclear environments. Finally, continuous studies on mRNA-mediated feedback mechanisms that slow down transcription to allow ordered processing and efficient export of mRNA into the cytoplasm are anticipated to shed light on the function of enhancers and the interconnection of nuclear processes.

Elucidating the mechanism(s) of enhancer function is not only relevant to understanding principles of gene regulation but also important for understanding the cause and progression of diseases. There are many diseases, including cancer, that are caused by mutations in noncoding regions that positively or negatively affect the function of enhancers/super-enhancers [26,83]. For example, an insertion of a small DNA sequence that recruits transcription factor Myb upstream of the Tal1 gene locus creates a super-enhancer that drives high level Tal1 expression in T-cell acute lymphoblastic leukemia [84,85]. Another example is the mutation of a transcription factor GATA1-binding site in an erythroid enhancer driving expression of the repressor protein BCL11A [86,87]. Reduced transcription of BCL11A is associated with elevated expression of the fetal γ-globin gene. Patients with sickle cell anemia harboring this additional mutation exhibit milder symptoms. This shows that understanding how mutations affect enhancer function and disease progression may inform novel therapeutic strategies.

## Figures and Tables

**Figure 1 ijms-23-09290-f001:**
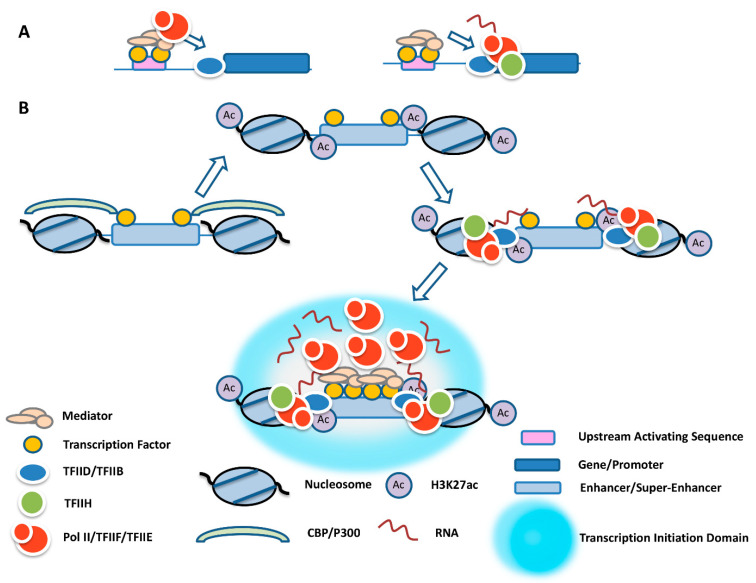
Formation of transcription initiation domains centered at enhancers. (**A**) Recruitment of an incomplete Pol II transcription complex to a yeast UAS. Pol II, together with TFIIF and TFIIE, is recruited to the UAS via Mediator through DNA sequence-specific transcription factors and transferred to TFIID-bound promoters, where it associates with TFIIH to initiate transcription. (**B**) The multistep process of transcription initiation domain formation at enhancers. Transcription factors at enhancers recruit CBP/p300, which acetylates flanking nucleosomes at H3K27. H3K27ac interacts with the TFIID complex via TAF1. TFIIB recruits Pol II and TFIIH, leading to transcription and the formation of eRNA. After release, eRNA, together with Mediator and other proteins, participates in the formation of a transcription initiation domain that concentrates Pol II.

**Figure 2 ijms-23-09290-f002:**
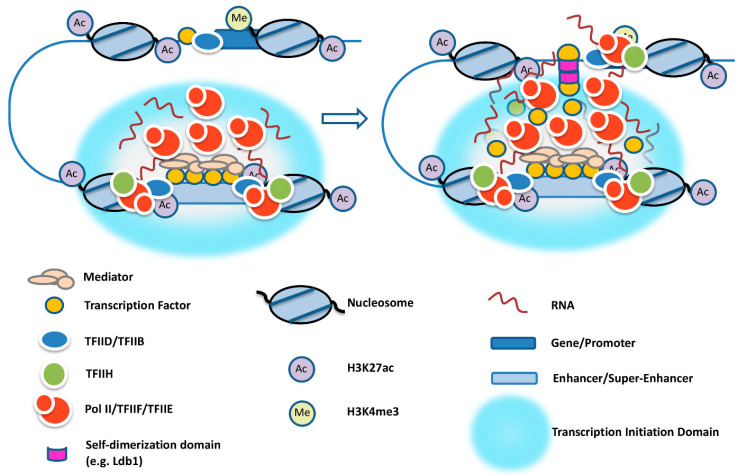
Establishing transient contacts of activated genes with enhancer or super-enhancer transcription initiation domains. Target genes are bound by TFIID/TFIIB, which is regulated by transcription factors (orange circle). Transient contacts between genes and enhancer-mediated transcription initiation domains are governed by transcription factors and coregulators like Ldb1 (purple disc) that are bound to promoters and tethered at the transcription initiation domains. The interaction of eRNA with DNA through triplex formation, or with promoter upstream transcripts (PROMPTs) through RNA binding proteins or RNA:RNA interactions, provides further energy and specificity for the transient interactions of the genes with the transcription initiation domains.

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
