# Peer review of "Enhancer-Mediated Formation of Nuclear Transcription Initiation Domains"

_ijms, 2022, doi:10.3390/ijms23169290_

Round 1
Reviewer 1 Report
This review is superb and timely. There is a thorough analysis of recent papers on the subject, many of which were just recently published. This review will be helpful to the field of transcription and will be well cited. There were a few minor points:
Line 45, awkward phase, “…regardless with respect to…” used in this sentence. May be better to state, “…independent of distance, position, and orientation.”
Line 82, should read “form” instead of “from.”
Line 147, “Taf1” should be capitalized. TAF1. This has been the convention throughout the paper to refer to the human proteins, which is appropriate.
Line 183, should read, “depleted”
Line 185, capitalize to read, “Drosophila.”
Line 186, awkward sentence, difficult to understand. May be better to state, “…half-life of paused Pol II…”
Line 188, should read, “half-life.”
Most figures were blurry in my rendition of the manuscript. Each structure looked fuzzy rather than crisp. I think sharper borders would be preferred and easier on readers. High resolution schematics will be preferred in the final manuscript. In Figure 2, self-dimerization domain is depicted in green, which makes it difficult to notice in the schematic since Pol II complexes are also shown in green. Consider changing color and size to make the point of the figure.
Author Response
This review is superb and timely. There is a thorough analysis of recent papers on the subject, many of which were just recently published. This review will be helpful to the field of transcription and will be well cited. There were a few minor points:
Line 45, awkward phase, “…regardless with respect to…” used in this sentence. May be better to state, “…independent of distance, position, and orientation.”
We changed this accordingly.
Line 82, should read “form” instead of “from.”
We changed this to “form”.
Line 147, “Taf1” should be capitalized. TAF1. This has been the convention throughout the paper to refer to the human proteins, which is appropriate.
We capitalized TAF1 throughout the text.
Line 183, should read, “depleted”
We changed this.
Line 185, capitalize to read, “Drosophila.”
We changed this.
Line 186, awkward sentence, difficult to understand. May be better to state, “…half-life of paused Pol II…”
We changed this accordingly..
Line 188, should read, “half-life.”
We changed this
Most figures were blurry in my rendition of the manuscript. Each structure looked fuzzy rather than crisp. I think sharper borders would be preferred and easier on readers. High resolution schematics will be preferred in the final manuscript. In Figure 2, self-dimerization domain is depicted in green, which makes it difficult to notice in the schematic since Pol II complexes are also shown in green. Consider changing color and size to make the point of the figure.
We changed the figures and avoided fuzzy structures. We also changed the color and size of the box depicting the self-dimerization domain. We provide high resolution images with the revised manuscript
Reviewer 2 Report
In this review, Gibbons et al discuss the formation of transcription initiation domains at enhancers and their role in recruiting Pol II to promoters. The article generally reads well for someone very well-versed in the field. However, I find the review to be a little disorganized and difficult to follow, especially for readers naïve to the field. Also, it reads as if the authors have just summarized a large number of facts without synthesizing or connecting different ideas in the field, organizing them and discussing unanswered questions.
Specific comments that may help to improve the manuscript are below.
Major comments:
1. Many terms need better explanation, and a general introduction to transcription initiation could help with this. Many ideas/terms are mentioned early and then introduced later. For instance, mediators, members of the transcription complex like TFIID etc. It might help if the authors could briefly introduce the Pol II transcription complex early and then talk in more detail about its recruitment to promoters, enhancers etc.
2. Instead of just listing recent discoveries, the authors should try to synthesize them into a coherent narrative to guide the readers through the logic of past work, why it is significant and what still needs to be answered.
3. Many statements lack supporting references. e.g., Line 135, 141, 156
4. The text seems very scattered and needs to be reorganized. Many components of the transcription complex are mentioned earlier but defined later. e.g., TFIID.
5. Instead of starting with a section on enhancers and super-enhancers with vague and unexplained terms, the authors could begin with a detailed section on transcription complex and promoters. This will make the review easier to understand, especially for the readers not well versed in the field.
6. The section with the recruitment of Pol II to promoters needs more explanation. How is TFIIE recruited? Surprisingly, RNA Pol II preinitiation complex has not been included in the text at all.
7. In the abstract, authors write, “We propose that there are two modes of Pol II recruitment to enhancers in higher eukaryotes”. However, this doesn’t seem to be the case. Rather than two different modes of Pol II recruitment, this seems to be a feed-forward loop where Pol II recruitment at enhancers leads to the generation of eRNA, which further promotes Pol II binding through the formation of a transcription initiation domain. The authors should rephrase their statement.
8. Figure 1A really doesn't seem to convey much information beyond what is said in the text and should be omitted.
9. Instead, the authors could add a new figure showing the differences as well as conserved features of Pol II recruitment by transcription factors and co-regulators at distal regulatory elements from yeast (via UASs) to higher eukaryotes (via enhancers).
10. Line 151: Better explanation is needed. Are the authors trying to say that Pol II binding at SEs is transient or dynamic?
11. Line 156: The authors should provide more explanation about why Pol II recruited by the other mode are transcriptionally incompetent.
12. Transcription bursting also happens in S. cerevisiae, which do not have enhancers. The authors should make this clear in the section “Enhancer mediated transcription bursting” that transcription bursting can happen without enhancer as well.
13. The section “The role of enhancer transcription and eRNA” is very poorly written. Re-writing with relevant information will help in getting the message across.
14. In the abstract, the authors mention the link between mis-regulation of enhancers and a variety of human diseases, including cancer. However, this has not been elaborated in the main text at all.
Minor comments:
1. Line 82. Do the authors mean “enhancers form phase-separated transcription initiation domains”?
Author Response
Reviewer 2:
In this review, Gibbons et al discuss the formation of transcription initiation domains at enhancers and their role in recruiting Pol II to promoters. The article generally reads well for someone very well-versed in the field. However, I find the review to be a little disorganized and difficult to follow, especially for readers naïve to the field. Also, it reads as if the authors have just summarized a large number of facts without synthesizing or connecting different ideas in the field, organizing them and discussing unanswered questions.
Specific comments that may help to improve the manuscript are below.
Major comments:
- Many terms need better explanation, and a general introduction to transcription initiation could help with this. Many ideas/terms are mentioned early and then introduced later. For instance, mediators, members of the transcription complex like TFIID etc. It might help if the authors could briefly introduce the Pol II transcription complex early and then talk in more detail about its recruitment to promoters, enhancers etc.
We moved up the section on recruitment of Pol II to promoters as per recommendation.
- Instead of just listing recent discoveries, the authors should try to synthesize them into a coherent narrative to guide the readers through the logic of past work, why it is significant and what still needs to be answered.
In some of the sections we list more recently published facts that we believe are relevant with respect to the model of Pol II recruitment and transfer. We attempted as much as possible to put these facts in the context of the main points raised in this review.
- Many statements lack supporting references. e.g., Line 135, 141, 156
These lines refer to speculations or possible scenarios based on facts referenced in other sections. We added a reference to line 141 (now line 145, after accept changes), reference 11.
- The text seems very scattered and needs to be reorganized. Many components of the transcription complex are mentioned earlier but defined later. e.g., TFIID.
This is related to point 1. We moved up the section on Pol II recruitment to the promoter, which now defines TFIID and other components of the transcription complex earlier.
- Instead of starting with a section on enhancers and super-enhancers with vague and unexplained terms, the authors could begin with a detailed section on transcription complex and promoters. This will make the review easier to understand, especially for the readers not well versed in the field.
This is again related to point1 . We moved this section up and expanded our description of the transcription complex.
- The section with the recruitment of Pol II to promoters needs more explanation. How is TFIIE recruited? Surprisingly, RNA Pol II preinitiation complex has not been included in the text at all.
We now include a description of TFIIE and the preinitiation complex in the section “Promoter and formation of a Pol II transcription complex”. (Lines 59-62, after accept changes))
- In the abstract, authors write, “We propose that there are two modes of Pol II recruitment to enhancers in higher eukaryotes”. However, this doesn’t seem to be the case. Rather than two different modes of Pol II recruitment, this seems to be a feed-forward loop where Pol II recruitment at enhancers leads to the generation of eRNA, which further promotes Pol II binding through the formation of a transcription initiation domain. The authors should rephrase their statement.
We agree with the reviewer and changed the abstract accordingly.
- Figure 1A really doesn't seem to convey much information beyond what is said in the text and should be omitted.
We agree that this figure does not convey much information. We slightly changed it to indicate that recruitment of transcription competent Pol II and TFIIH is restricted to the promoter in yeast. We would like to keep this figure (1A) to illustrate the difference between yeast and higher eukaryotes (as outlined in Figure 1B).
- Instead, the authors could add a new figure showing the differences as well as conserved features of Pol II recruitment by transcription factors and co-regulators at distal regulatory elements from yeast (via UASs) to higher eukaryotes (via enhancers).
We think that Figure 1 shows the difference and similarities in Pol II recruitment between yeast and higher eukaryotes.
- Line 151: Better explanation is needed. Are the authors trying to say that Pol II binding at SEs is transient or dynamic?
We changed this sentence according to the reviewer suggestions. We believe that enhancers/super-enhancers dynamically and transiently recruit Pol II, which is available for transfer to promoters.
- Line 156: The authors should provide more explanation about why Pol II recruited by the other mode are transcriptionally incompetent.
We expanded on this in the revised manuscript. Specifically, we suggest that most Pol II recruited to enhancers does not engage in transcription but is loosely bound to co-activators in the context of the transcription initiation domain and then transferred to TFIID-bound promoters during transient contacts. See last paragraph of page 4 (after accept changes).
- Transcription bursting also happens in S. cerevisiae, which do not have enhancers. The authors should make this clear in the section “Enhancer mediated transcription bursting” that transcription bursting can happen without enhancer as well.
We appreciate this comment and discuss bursting in yeast in the revised manuscript. (Lines 178-180, after accept changes)
- The section “The role of enhancer transcription and eRNA” is very poorly written. Re-writing with relevant information will help in getting the message across.
We changed this section, including addition of introductory and summarizing paragraphs to highlight the main points. See pages 5 and 6 (after accept changes)
- In the abstract, the authors mention the link between mis-regulation of enhancers and a variety of human diseases, including cancer. However, this has not been elaborated in the main text at all.
We removed the sentence from the abstract and added a new paragraph at the end of the manuscript (page 10) in which we highlight the importance of understanding enhancer function with regard to increasing evidence of disease causing mutations in enhancers/super-enhancers. This is not a focus of our review and we include references that summarize these important findings.
Minor comments:
- Line 82. Do the authors mean “enhancers form phase-separated transcription initiation domains”?
We changed this.
Reviewer 3 Report
Summary
This manuscript is a review of enhancer function in eukaryotes, with a primary focus on metazoans. In the abstract, the authors propose two modes of PolII recruitment to enhancers, one involving TFIID and another involving eRNAs. There is a brief introduction followed by sections on enhancers/super enhancers, promoters and transcription initiation, PolII recruitment to enhancers, and transcriptional bursting. There is then a long section on enhancer function and eRNAs.
General comments
The topic of the review is of broad interest and is a very active area of investigation. The thesis introduced in the abstract is an interesting perspective on an enigmatic feature of enhancer function. However, there are some structural and organizational issues with the review that limited my enthusiasm. The proposal advanced in the abstract regarding two modes of PolII recruitment to enhancers is not explored in detail in subsequent sections and is not really the focus of the review. It is not mentioned in the introductory section. There is a short section devoted specifically to PolII recruitment to enhancers in which the two modes of recruitment are introduced again. Here the issue is framed slightly differently, implying that one mode involves productive transcription whereas the other does not. This is interesting but also confusing, as it is unclear how it relates to what was in the abstract. There was also no literature presented to support the existence of these two modes. The rest of the review focused largely on enhancer transcription/eRNAs. I think that a re-focusing on providing evidence for these two modes of PolII recruitment to enhancers, and decreased focus on eRNAs, would improve the flow and readability. It would also help to distinguish this review from many others that have appeared recently (eg ref 12, 25).
Specific comments
Line 53: what is the evidence for enrichment of particular histone variants at enhancers (no reference is provided)?
Line 119: what is the evidence that promoter histone acetylation and methylation are of primary importance in TFIID recruitment or positioning? These chromatin marks recruit many other factors, and TFIID recognizes promoter DNA
Line 137: Please clarify how the yeast paradigm for PolII recruitment to a UAS informs the situation in metazoans. Is this an example of a non-productive recruitment mode? Seems like it from ref 10 but it is not spelled out clearly. These references also seem relevant to the yeast paradigm: 10.1038/nsmb.2810, 10.1016/j.molcel.2014.03.024
Line 210: what is the evidence that PolII elongation-associated histone marks and/or CTD ser2 phosphorylation are present at enhancers? This paragraph seems to suggest that they are.
Line 233: Is enhancer transcription uni- or bi-directional? This needs to be explained more clearly. Also, the comments about R-loops regulating directionality need to be explained more fully to be appreciated by a general audience. No background on R-loops is provided.
Figures: what is the difference between 1B and 2?
Author Response
General comments
The topic of the review is of broad interest and is a very active area of investigation. The thesis introduced in the abstract is an interesting perspective on an enigmatic feature of enhancer function. However, there are some structural and organizational issues with the review that limited my enthusiasm. The proposal advanced in the abstract regarding two modes of PolII recruitment to enhancers is not explored in detail in subsequent sections and is not really the focus of the review. It is not mentioned in the introductory section. There is a short section devoted specifically to PolII recruitment to enhancers in which the two modes of recruitment are introduced again. Here the issue is framed slightly differently, implying that one mode involves productive transcription whereas the other does not. This is interesting but also confusing, as it is unclear how it relates to what was in the abstract. There was also no literature presented to support the existence of these two modes. The rest of the review focused largely on enhancer transcription/eRNAs. I think that a re-focusing on providing evidence for these two modes of PolII recruitment to enhancers, and decreased focus on eRNAs, would improve the flow and readability. It would also help to distinguish this review from many others that have appeared recently (eg ref 12, 25).
We do not agree with this reviewer. The focus of this review is on the different mechanisms by which Pol II is recruited to enhancers/super-enhancers and how transient contacts with promoters are established. We emphasize on this throughout the review and both of the figures focus on the modes of Pol II recruitment and establishment of promoter-enhancer contacts.
Specific comments
Line 53: what is the evidence for enrichment of particular histone variants at enhancers (no reference is provided)?
Many ChIP-seq data reveal increased levels of H2K27ac and H3K4me at enhancers. This has been reviewed before and we refer to these reviews in our article.
Line 119: what is the evidence that promoter histone acetylation and methylation are of primary importance in TFIID recruitment or positioning? These chromatin marks recruit many other factors, and TFIID recognizes promoter DNA
It has been established that TAF1 recognizes acetylated histones and TAF3 recognizes H3K4 methylated histones. This has been reviewed by Marc Timmers and colleagues and we cite this review in our article (Bhuiyan and Timmers, reference 17 in the revised manuscript).
Line 137: Please clarify how the yeast paradigm for PolII recruitment to a UAS informs the situation in metazoans. Is this an example of a non-productive recruitment mode? Seems like it from ref 10 but it is not spelled out clearly. These references also seem relevant to the yeast paradigm: 10.1038/nsmb.2810, 10.1016/j.molcel.2014.03.024
The findings in yeast and higher eukaryotes suggest that recruitment of Pol II to upstream activating sequences and enhancers is a conserved process. It also suggests that the Pol II transfer mechanism is conserved.
Line 210: what is the evidence that PolII elongation-associated histone marks and/or CTD ser2 phosphorylation are present at enhancers? This paragraph seems to suggest that they are.
ChiP-seq data show that Pol II, phosphorylated at serine 2, is found at enhancers. For example, see Koch et al. ( reference 10 in our revised manuscript). The level of serine 2 phosphorylation is lower at enhancers. This is consistent with our hypothesis that most Pol II complexes recruited to enhancers are not engaged in transcription.
Line 233: Is enhancer transcription uni- or bi-directional? This needs to be explained more clearly. Also, the comments about R-loops regulating directionality need to be explained more fully to be appreciated by a general audience. No background on R-loops is provided.
We discuss this in our review. There is bidirectional transcription, however, recent data suggest that this rarely happens in a single cell. We cite a recent study from the Proudfoot lab showing that R-loops promote bidirectional transcription.
Figures: what is the difference between 1B and 2?
Figure 1 focuses on the mechanisms of Pol II recruitment to enhancers and UASs, while Figure 2 focuses on how transient contacts with promoters are established.
Round 2
Reviewer 2 Report
The authors have addressed all my concerns in the revised manuscript.
Author Response
We thank the reviewer for the constructive comments.
Reviewer 3 Report
The authors have improved the flow and organization in their revised version. There is also a clearer explanation of their model, which proposes two modes of PolII recruitment to enhancers. However, it still seems to me that much of the information after section 4 takes the focus away from this model. Some text should be included to link this information back to their proposed model.
Author Response
We thank the reviewer for the constructive comments on our manuscript. We added new text (highlighted in yellow) to the Introduction, lines 401 and 402; to the beginning of section 6, lines 204 to 206; and to the end of section 6, lines 317-321.